# Bubble in Carbon Credits during COVID-19: Financial Instability or Positive Impact ("Minsky" or "Social")?

Bikramaditya Ghosh [1], Spyros Papathanasiou [2,*], Vandita Dar [1] and Konstantinos Gravas [3]

1    Symbiosis Institute of Business Management, Symbiosis International (Deemed University) Electronic City, Hosur Road, Bengaluru 560100, India
2    Department of Economics, School of Economics and Political Sciences, National and Kapodistrian University of Athens, 1 Sofokleous Street, 10559 Athens, Greece
3    Department of Banking and Financial Management, School of Finance and Statistics, University of Piraeus, 80 M. Karaoli & A. Dimitriou Street, 18534 Piraeus, Greece
*    Correspondence: spapathan@econ.uoa.gr

**Abstract:** Incentivizing businesses to lower carbon emissions and trade back excess carbon allowances paved the way for rapid growth in carbon credit ETFs. The use of carbon allowances as a hedging alternative fueled this rally further, causing a shift to speculation and forming repetitive bubbles. Speculative bubbles are born from euphoria, yet, they are relatively predictable, provided their pattern matches the log periodic power law (LPPL) with specific stylized facts. A "Minsky moment" identifies a clear speculative bubble as a signal of financial system instability, while a "Social bubble" is regarded as relatively positive, increasing in the long run, infrastructure spending and development. The aim of this paper is to investigate whether various carbon credit bubbles during the pandemic period caused financial instability or had a positive impact ("Minsky" or "Social"). Particularly, we investigate the carbon credit bubble behavior in the ETF prices of KRBN, GRN (Global Carbon Credit tracking ETFs), and the SOLCARBT index during the COVID-19 pandemic period by adopting the log-periodic power law model (LPPL) methodology, which has been widely used, over the past decade, for detecting bubbles and crashes in various markets. In conclusion, these bubbles are social and propelled by the newfound interest in carbon credit trading, for obvious reasons.

**Keywords:** green finance; green energy; COVID-19; carbon credit; log periodic power law; social bubble hypothesis; COVID-19

## 1. Introduction

According to Minsky (2016), "the essential financial process of a capitalist economy center around the way investment and positions in capital assets are financed". Furthermore, the stability of an economy's financial structure depends upon the mix of three financial postures for firms, households, and government units, which can be differentiated by the relation between the contractual payment commitments due to their liabilities and their primary cash flows. These financial postures are hedge, speculative, and "Ponzi".

Minsky (2016) defines a Ponzi finance unit as "a speculative financing unit for which the income component of the near term cash flows falls short of the near term interest payments on debt so that for some time in the future the outstanding debt will grow due to interest on existing debt". Ponzi schemes have an impact on both the financial system and society (Frankel 2012). In this regard, a "Minsky moment" identifies a clear speculative bubble as a signal of financial system instability.

On the other hand, some bubbles may even have positive social consequences. For example, several economically beneficial technologies would not have been developed without the assistance of bubbles. A "Social bubble" is regarded as relatively positive in the long run regarding economic development. (Quinn and Turner 2020).

The carbon credit transfer was born through incentivizing businesses to pollute less, on the one hand, while still making profits during the process, on the other. This is possible via the "cap and trade" system, using Carbon Credit ETFs (Exchange Traded Funds), such as KRBN (Krane Shares Global Carbon Strategy ETF), and GRN (iPath Series B Carbon ETN), which mimic the returns of carbon allowances in this system. Moreover, as the prices of these instruments have a very low correlation with stocks, commodities, and bonds, like other commodities, they can be used as a possible hedge (Samitas et al. 2022a).

The government incentivizes businesses to reduce carbon emissions. Carbon pricing is implemented as a means of reducing Greenhouse Gas Emissions (GHG emissions) either in form of a carbon tax or by buying the permits to pollute (Sahu and Patnaik 2020). This permit to pollute is commonly referred to as a "cap and trade" or carbon credit trading/exchange scheme. Under the "cap and trade" system, governments limit even specific allowances for the owner of the permit. If they need to emit more GHG, they have to buy additional allowances, which are typically determined by the open market (Ghosh et al. 2022a). To add to the same debate, global investment funds are chasing after environmental, social, and governance principles (ESG) (Umar et al. 2020).

KRBN and GRN typically own the shares of carbon ETFs. Usually, an investor of these two ETFs owns carbon allowances, which other businesses are ready to purchase. Since the supply is limited, buying would drive the demand higher. Certain businesses cannot generate revenue without the emission of GHG (Le et al. 2020; Ramirez-Contreras et al. 2020; Leal et al. 2019). Thus, it becomes essential for them to buy the carbon rights allowances, regardless of the price. High pollution businesses find these carbon allowances essential for their business.

Most companies that are in energy transition mode sometimes have an excess of carbon allowances. Hence, they trade them to high polluting companies. Trading carbon allowances for profit comes as a reward for polluting less. Since these ETFs buy excess carbon allowances beforehand from the relatively low polluting companies, they make a substantial profit in selling them back to high polluting companies. Moreover, governments issue fewer carbon allowances each year in comparison to the previous year, thus reducing their supply even further. This too has a substantial impact on ETF prices, which surge quite fast, hence even generating a speculative bubble at times. Forward projections of carbon allowances for 2020 were $30, which was quite accurate. However, further growth potential is projected to be to the tune of 3–4 X. John Kerry, as the special advisor behind the index of carbon ETFs, has predicted a price of $100 per tonne[1]. However, after joining the Biden administration as a US special presidential envoy for climate, he revised his prediction to about $51 per tonne. At a later stage, this projection was further revised to $125 per tonne. Ricke et al. (2018), a famous carbon scientist, takes the projection for prices of sustainable carbon allowances further to $150–200 per tonne (Ricke et al. 2018).

Currently, the KRBN universe covers California, New England, and the EU, with the potential for more. Sweden has a target price of about $126 per tonne[2], which is not considered extremely aggressive, as they had priced them around $20 in 1991 hence a growth of little more than 6% per year. In fact, carbon allowances in the EU were the best-performing global commodity for some time now (2018 and 2019). According to the World Bank, 21.5% of global GHG emissions would be covered by carbon credits through "cap and trade"[3]. Moreover, carbon ETFs such as KRBN and GRN could also act as a hedge against losses from fossil fuel companies (Samitas et al. 2022b). Further, they can also hedge against credit downgrades and environmental lawsuits against high-carbon-emitting companies.

Carbon allowances hardly have any significant correlation with traditional assets such as equities, bonds, commodities, real estate, and even gold (Zhang et al. 2017). This indicates possible outperformance and diversity for KRBN and GRN ETFs. Therefore, offsetting the carbon footprint through the purchase of carbon credits has witnessed a rise of late. The beneficial role of carbon offsets is advertised like never before[4] (Koutsokostas et al. 2019; Koutsokostas et al. 2018; Christopoulos et al. 2014; Filimonov and Sornette 2013). Strict commitments to zero out emissions within a short timeframe by purchasing offsets is

definitely one of the solutions; perhaps, a convenient one. Reducing the carbon footprint remains the global motto. Having said that, participating in this through the transfer of carbon credit could well appear to be a quick fix rather than actually cutting emissions.

The aim of this paper is to investigate whether various carbon credit bubbles during the pandemic period caused financial instability or had a positive impact ("Minsky" or "Social"). Particularly, we investigate the carbon credit bubble behavior in the ETF prices of KRBN, GRN (Global Carbon Credit tracking ETFs), and the SOLCARBT index during the COVID-19 pandemic period, by adopting the log-periodic power law model (LPPL) methodology, which has been widely used over the past decade. So, in this study, we investigate the carbon credit bubble in the prices of ETFs using KRBN, GRN (Global Carbon Credit tracking ETFs), and the SOLCARBT index during COVID-19. First of all, this study found 13 carbon credit bubbles during the COVID-19 period. All the carbon credit bubbles followed the LPPL signature when re-calibrated using the Filimonov and Sornette (2013) LPPL equation. Secondly, the average drawdown observed was two times that of S&P-500 listed stocks (54% vs. 27%) under trying conditions, such as COVID-19.

An exchange-traded fund (ETF) is a type of pooled investment security that operates much like a mutual fund. Typically, ETFs will track a particular index, sector, commodity, or other asset. ETFs can be purchased or sold on a stock exchange the same way that a regular stock can.

In order to detect carbon credit bubbles, the log-periodic power law model was developed (Sornette et al. 1996; Drozdz et al. 1999). The aim of the log-periodic power law model is to investigate whether the LPPL patterns in the development of carbon credit bubbles can be conducive to default classification.

Over the past decade, the LPPL model has been widely used for detecting bubbles and crashes in various markets (Zhou et al. 2018; Li 2017; Wątorek et al. 2016). An important advantage of the LPPL model relative to other approaches is that it seeks to predict both the continuation and termination of a bubble in the same estimation. The notion that financial crashes are manifestations of power law accelerations essentially suggests that endogenously induced carbon market crashes might obey a particular power law, with log-periodic fluctuations (Brée and Joseph 2013).

This study contributes to the existing literature as it searches for a common thread across the carbon credit bubble of the prices of two ETFs during COVID-19 by using the Filimonov and Sornette (2013) modified LPPL: (GRN (iPath Series B Carbon ETN) which holds futures contracts on EUAs (European Union Allowances) and CERs (Certified Emission Reduction) that measures the performance of emissions units and KRBN (Krane Shares Global Carbon Strategy ETF), which is benchmarked to IHS Markit's Global Carbon Index, which offers broad coverage of cap-and-trade carbon allowances by tracking the most traded carbon credit futures contracts) and the SOLCARBT index (Solactive Carbon Emission Allowances Rolling Futures Index).

This study is significant for two reasons: Firstly, in contrast to prior relevant studies, it examines whole baskets of the most traded carbon credit futures contracts (GRN and KRBN ETFs) and the SOLCARBT index (Solactive Carbon Emission Allowances Rolling Futures Index) during COVID-19, from September 2019 to November 2021 by using the Filimonov and Sornette (2013) modified LPPL. In addition, we differentiated from previous studies (Geuder et al. 2019; Ghosh et al. 2021, 2020, 2022b; Kenourgios et al. 2021; Wheatley et al. 2019) by testing the robustness of the LPPL following the reformulated version of the LPPL calibrations proposed by Filimonov and Sornette (2013). Secondly, policymakers would benefit from this research, as they will redefine strategies and apply targeted climate policy measures following bio-economy routes to sustainable, post-GHG societies. They will improve the efficiency of industrial production and enhance renewable energy technology.

This study is composed of five sections. Section 2 presents the literature review. Section 3 describes the data and the methodology used. In Section 4, the outcomes and findings of the research are presented. Finally, in Section 5, the conclusions are reported.

## 2. Literature Review

Bubbles are generally seen in a negative light, as transient, short-lived, encompassing uncertainty, causing crises and painful downturns.

Vliamos and Gravas (2018) mention Akerlof's and Schiller's interpretation of Keynesian animal spirits; "as in the level of macroeconomics, overall, confidence comes and passes, it is not simply a rational prediction but the first and most important of our animal spirits". In the same sense, drawing from behavioral economics, bubbles can largely be explained by cognitive failings and psychological biases on a subset of investors suffering, for example, "from an overconfidence bias" (Quinn and Turner 2020).

Interestingly, Minsky's seminal "financial instability hypothesis" as a much more plausible explanation of a capitalist economy followed Keynes' rebuttal to Jacob Viner's critical review of his General Theory (Minsky 1977). Keynes' elegant explanation of financial and output instability within a capitalist set-up—often obscured by its predecessor, the neoclassical theory—was brought out sharply in his incisive response to Viner's criticism. In fact, Minsky attributed the historically observed recurring unstable behavior as being inherent in the capitalist system. He put forth his hypothesis "not as an interpretation of Keynes but rather as an alternative to the current, standard neoclassical theory" (Minsky 2016). The structural Keynesian view of Palley (2009) suggested that Minsky's financial instability hypothesis also proffered a convincing argument for the post-1980 neo-liberal growth model, which was primarily driven by debt and asset price inflation, rather than an increase in aggregate demand or growth in the real economy. Minsky traced the evolving inherent systemic financial instability as it transforms from a "basic Minsky cycle" to a "super-Minsky cycle"; beginning with "hedge finance", characterized by conservatism when borrowers' revenues are sufficient to cover interest as well as principal repayments; then, transitioning to "speculative finance", where revenues might cover just the interest payments. Eventually, an irrational euphoria on the part of market participants leads to the "Ponzi finance" phase, where borrowers expect to meet their obligations through capital gains, while revenues fall short of cash payment commitments (Palley 2009; Keen 1995). As Minsky put it, "the mix of hedge, speculative, and Ponzi finance in existence at any time reflects the history of the economy and the effect of historical developments upon the state of long-term expectations" (Minsky 2016). Furthermore, the above process is aided by an environment of regulatory laxity; a three-dimensional phenomenon encompassing, first, regulatory capture, which is reflected in a weakening of regulatory institutions, second, regulatory relapse, where regulators are influenced by memory loss often justified by newer ideological grounds and third, regulatory escape, when current regulation is inadequate for innovative, complex financial products (Palley 2009). Eventually, unsustainable debt to equity ratios and concomitantly surging interest rates render many businesses unviable, resulting in a collapse of asset prices and the system imploding.

A unique perspective of viewing a bubble was popularized by Garber (1990), who suggested that apparently speculative events could frequently be reasonably explained by economic fundamentals and sound financial logic, such as expectations of high returns. Some bubbles could result in positive social and economic outcomes and trends such as spurring entrepreneurs, augmenting innovation as well as attracting capital (Quinn and Turner 2020). The social bubble hypothesis was evident in the success of the Human Genome Project (HGP), and in fact, it could be playing out in other critical sectors, such as clean or renewable energy and biotechnology, attracting capital and resulting in positive collaborative networks between private and public stakeholders (Gisler et al. 2011; Mathews 2013; Knuth 2018; Papathanasiou et al. 2022; Marcus et al. 2013). The challenge here is that most of the time, the positive impact of a social bubble is not immediately visible, while the crash stemming from a bubble implosion is obviously evident at the moment it happens. Hence classifying a speculative episode as a "Minsky moment" without due recognition of it is basic character could be misleading.

Globally, carbon markets emerged during the 1990s as a concerted policy effort to tackle climate change. Carbon trading was viewed as the more efficient alternative to taxing

polluting entities or coercively imposing limits on emissions, which is characteristic of a "command and control" approach (Milner 2007). Policymakers fervently hoped that carbon trading would help finance clean technologies in poorer countries with higher emissions, necessitated by concerns about economic growth.

However, the evolution of the carbon credit markets can be traced back to the 1960s, when it was first proposed by policy-makers in the US as pollution trading and later it was further developed by economists and traders in the 1970s (Lohmann 2010). It culminated as the cornerstone of the US Acid Rain Programme in the 1990s after a series of failed policy experiments. Eventually, the Kyoto Protocol of 1997 recognized greenhouse gas emissions as a new commodity, based on countries' accepted targets for reduced emissions. The permissible emissions were divided into assigned amount units (AAUs), and countries that polluted less than their prescribed limit could sell this excess capacity to countries that overshot their targets (UNFCC 1997)[5]. The adopted nomenclature "carbon market", stemmed from the fact that carbon dioxide was predominant when it came to greenhouse gas emissions. Viewing carbon credits as potent instruments for laying the foundation of a sustainable, renewables-based energy system, Mathews (2008) envisaged future investment and economic decisions largely dictated by the pricing of carbon. Other than carbon taxes and cap-and-trade schemes, he suggested an alternative approach where credible private sector financial institutions could issue tradable carbon sequestration certificates. Besides government and global regulation that would support the demand for carbon credits, competitive pressures and the pricing of carbon would further incentivize the creation of a much larger Emissions Trading System (ETS).

Carbon trading understandably has its fair share of critics. Lohmann (2008) offers some extremely valid points to ponder about. The preoccupation with carbon dioxide cuts that accompany carbon credits might actually be making costly trade-offs with other equally harmful pollutants; in addition, the market is designed to dilute the government's role in regulation, as corporations can buy the right to pollute. Moreover, there is a blatant lack of attention paid to where and how the cuts are made, breeding inequalities and working against climate justice. The proposed equivalence between carbon credits emanating from emission allowances and offset projects for convenience need not be an efficient approach, both differing in terms of the course pursued. Carbon markets are also propagating a dangerous homogeneity in environmental preservation, cloaked in jargon and a technocratic approach that is characteristic of carbon markets, often undermining the traditional and valuable local knowledge base that exists around the world.

Pearse and Böhm (2014) highlight multiple stakeholders' view of carbon markets propagating "climate injustice", as these markets have not succeeded in reducing emissions, while at the same time exacerbating inequalities and unequal development. This is because carbon markets have enabled developed countries to reduce emissions largely through offset projects in middle-income, developing countries, which in many cases did not entail real carbon reductions. Furthermore, the low-income, poorer households suffered the most when the cap-and-trade mechanism was applied to fossil fuel industries, impacting energy prices and prices of all other goods and services.

Michaelowa et al. (2019) have traced the evolution and development of carbon markets right from their emergence and operation in the 1990s to the golden period between 2005–2011, followed by a lack of demand and fragmentation till about 2014. The Post-Paris-Agreement phase, from 2015 onwards, is marked by increasing efforts toward global participation, which while well-intentioned appear ambitious and fraught with challenges, such as demand-supply balance; environmental integrity; costs, as well as conflicts with national policies and targets.

Woo et al. (2021) have found that the digital MRV (Measurement, Reporting, and Verification) system, which is required by climate action projects can be applied to the building sector with the adoption of Blockchain technology. Anjos et al. (2022) have studied the potential role and advantages of a multinational carbon-credit (CC) market allowing a set of countries to procure CCs for their domestic producers, thus observing the

varying effects of such a common CC market on government policy and the behavior of energy producers.

While 192 states in the world are in some form of agreement with the Kyoto Protocol, so far the signatories are only 83 (82 countries and the European Union). About 40 countries in the world, and over 20 cities, states, and provinces are using some carbon pricing mechanism, in the form of an Emissions Trading System (ETS), which is nothing more than carbon trading or a carbon tax that penalizes greenhouse gas emissions, which is precisely the carbon content of fossil fuels (World Bank)[6]. Carbon pricing mechanisms constitute about 13% of annual global greenhouse gas emissions. While this may appear to be a modest achievement given the alarming pace of environmental degradation currently witnessed globally, it is heartening to note that the global carbon market stood at $272 billion in 2020, expanding fivefold since 2017[7]. The development of successful carbon ETF markets since 2019, such as GRN and KRBN, lends credence to the fact that this market holds a lot of promise for investors, companies, and governments alike.

## 3. Data and Research Methodology

Time series of all kinds were studied through fractal lenses in the past. Most studies have confirmed financial time series to be fractal. Fractal properties of a time series mostly exhibit a "fractal tree", a Hierarchical Model or HM (Sornette and Johansen 1998). It has further sub-trees, with different scaling properties.

The Log Periodic Power Law or LPPL is nothing but a micro-state hierarchical fractal construct. Though it is conceived and germinated from a micro-state, it typically provides a macro-state understanding. It has been proved in the past that recurring positive feedback by the financial market participants (agents) often creates an upward spiral, which eventually propels the underlying time series to reach an extreme valuation limit within a short time. This phenomenon is usually coupled with a phase transition or a sudden crash. The growth phase and the decline phase individually are a mean-shifting process at a first glance; however, their increments are mean-reverting in nature. Moreover, these two phases individually have a long-range dependence or long memory. This bubble build-up phase with an upward spiral and sudden crash phase has a fractal premise (Wu 2012).

Bubble prediction using the LPPL dates back to 2000 when the Johansen et al. (2000) construct was famously introduced (Johansen et al. 2000). The group took into consideration certain assumptions such as trader's affinity, no holding of shares other than buy-sell, herding, etc. The LPPL is thus set on the premise of a feedback-driven herd, generating a speculative bubble followed by an eventual crash. Due to the complex nature of any complex communication-based system, it is rather difficult to put forth such a straightforward construct. These shortcomings sometimes generate a "false bubble alarm" in the LPPL. Nevertheless, it has been repeatedly proven to be the most consistent bubble prediction model, especially so after the introduction of the reformulated LPPL in 2013 by Filimonov and Sornette (2013). This reduced non-linear parameters and increased linear parameters in the said model to make it more robust.

We investigated KraneShares Global Carbon ETF (KRBN) and iPath Series B Carbon ETN (GRN) from the reformulated LPPL viewpoint. KRBN ETF tracks the IHS Markit Global Carbon Index; whereas GRN tracks the Barclays Global Carbon II Index. Hence, they provide perfect proxies for carbon credit or carbon allowances. There are in total four ETFs in this segment. Having said that, the two newer ETFs (KraneShares European Carbon Allowance ETF or KEUA and KraneShares California Carbon Allowance ETF or KCCA) began operating only in October 2021. This explains our rationale to study KRBN and GRN. We wanted to cover the CARP ETF launched by Wisdom Tree, tracking Solactive Carbon Emission Allowances Rolling Futures Total Return Index (SOLCARBT). However, we have to directly consider SOLCARBT, as CARP started around late August 2021. All data were acquired from Bloomberg and Thomson Reuters Datastream. Finally, our chosen proxies for tracking carbon allowances or carbon credit trading are:

1.　　KRBN (tracking IHS Markit Global Carbon),
2.　　GRN (tracking Barclays Global Carbon II TR) and
3.　　Solactive Carbon Emission Allowances Rolling Futures TR (SOLCARBT).

Our observation ranged from 10th September 2019 to 11th November 2021. This time range typically covers the COVID-19 pandemic.

The reformulated LPPL model by Filimonov and Sornette (2013) has been rather efficient in finding bubbles in diverse asset classes. This study has been carried out by the recalibrated Filimonov and Sornette (2013). It all nevertheless started with the Johansen et al. (2000) model, having more non-linear parameters:

$$y_t = A + B\,(t_c - t)^\beta + C(t_c - t)^{\beta}\cos\,(\omega\log(t_c - t)) + \phi \tag{1}$$

where $t_c$ is nothing but the day of the phase shift (bubble to crash); $\beta$ represents the exponent of exponential growth during both the bubble and crash phases; $y_t$ is the expected value of the logarithm of the price ($y_t > 0$); $\omega$ represents the angular magnitude of the oscillation during the bubble formation less structural information, albeit they are coefficients. $A$ with a condition $A > 0$, signifies a bias term that can be ignored when prices are normalized. Moreover, $A$ is the price at the peak of the bubble phase; $t$ is any time into the bubble preceding ($t < t_c$). $A$, $B$, $C$, and $\phi$ are units having. $B$ signifies the height of the bubble just before the inevitable crash ($B < 0$). $C$ is the magnitude of the oscillations around the exponential growth ($|C| < 1$). $\Phi$ is the phase shift parameter, from bubble to crash.

Quick movements along the mean shifting line, with the angular frequency ($\omega$), generate the exponent of bubble formation ($\beta$). Intuitively it gets generated by countless recurring loops of optimism and positive feedback from the market participants till the critical point ($t_c$). Furthermore, the Johansen et al. (2000) model had too many non-linear parameters, too many local minima, and was not in sync with back-propagation algorithms. Filimonov and Sornette (2013) made it possible post the crucial recalibration.

Therefore, Filimonov and Sornette (2013) amended the Johansen et al. (2000) construct model with less non-linear parameters:

$$y_t = A + B\,(t_c - t)\beta + C_1\,(t_c - t)\beta\cos\,(\omega log\,(t_c - t)) + C_2\,(t_c - t)\beta\sin\,(\omega log\,(t_c - t)) \tag{2}$$

where $C_1 = CCos\emptyset C_2 = Csin\emptyset$.

The Filimonov and Sornette (2013) has four linear variables ($A$, $B$, $C_1$, $C_2$) and three non-linear variables ($t_c$, $\omega$, $\beta$). The linear parameters ($A$, $B$, $C_1$, $C_2$) are based on the 'Standard slaving principle'. This indicates how multiple microstates can construct a macrostate. The slaving principle was introduced by Haken (1975), in order to understand complex macrostates, as a combination of many tiny microstates (Haken 1975). Since the birth of the stylized fact, that "bubbles are a combination of self-organized non-linear microstates having time-varying information flow", the slaving principle has been used. The function of the subordination procedure is to significantly reduce local minima in both two-dimensional space ($\beta$, $\omega$) and one-dimensional space ($t_c$). The NelderMead simplex construct is used for finding local minima in a multidimensional space.

The Filimonov and Sornette (2013) model, however, is quite sensitive to the input values of the LPPL. The accuracy of a bubble indicator usually depends upon the length of the input. The optimum length is from 5 trading days to 750 trading days. The Recalibrated (Filimonov and Sornette 2013) LPPL works accurately well within these constraints. Therefore, unlike investors, day traders do not usually find it useful.

## 4. Empirical Results

Table 1 presents the conditions of the LPPL parameters, stated in the literature review. We have to check whether or not all 13 crashes in the carbon credit bubble fit in at the same set of values declared in Table 1. Models built are showcased for $\beta = 0.33 \pm 0.18$, $\omega = 6.36 \pm 1.56$ and $\varphi = 0$ to $2\pi$. Drawdowns were calculated using the Filimonov and Sornette (2013) method "price coarse graining" algorithm with $\varepsilon = 0^8$.

**Table 1.** Stylized facts of the LPPL.

| Parameter | Constraint | Literature |
|:---:|:---:|:---:|
| $A$ | (>0) | Korzeniowski and Kuropka (2013) |
| $B$ | (<0) | Lin et al. (2014) |
| $C_1$ | (Cos function) | Filimonov and Sornette (2013) |
| $C_2$ | (Sine function) | Filimonov and Sornette (2013) |
| $t_c$ | (*t* to ∞) | Korzeniowski and Kuropka (2013) |
| $B$ | (0.1 to 0.9) | Lin et al. (2014) |
| $\Omega$ | (4.8 to 13) | Johansen (2003) |

Note: The table above exhibits the conditions of the LPPL parameters of Equation (2) used for empirical analysis.

Table 2 presents the coefficients of the LPPL parameters in Equation (2). Drawdown (DD) is the break between the local minima to the next local maxima and it is ≥14%. Table 3 represents the identified events behind seven bubble crashes (>25%). A significant LPPL signature is exhibited by all 10 bubble crashes in both KRBN, GRN ETFs, and SOLCARBT index from 10th September 2019 to 11th November 2021 (overlapping with the COVID-19 period). No false positive alarm was detected during the empirical tests. Extremely lower values of Root Mean Square Errors (RMSE) in all the 13 instances support the reformulated LPPL fitment. All the 13 past crash instances in KRBN, GRN ETF, and SOLCARBT index occurred with the following stylized facts:

1. $\beta = 0.52 \pm 0.38$
2. $\omega = 9.65 \pm 3.39$
3. Minimum Drawdown (%) = 14%
4. Average Drawdown (%) = 54%

**Table 2.** Coefficients of the LPPL parameters having a drawdown of >14%.

| Corporate/ Index | Bubble | Time | $t_c$ | $A$ | $B$ | $C_1$ | $C_2$ | $\beta$ | $\omega$ | DD (%) |
|:---:|:---:|:---:|:---:|:---:|:---:|:---:|:---:|:---:|:---:|:---:|
| KRBN | B1 | 2 November 2020 to 17 December 2020 | 33 | 3.30 | −0.01 | 0.00 | 0.00 | 0.86 | 9.92 | 28% |
| | B2 | 15 January 2021 to 16 February 2021 | 26 | 3.51 | −7.77 | 0.00 | 0.00 | 0.88 | 6.76 | 19% |
| | B3 | 17 March 2021 to 14 May 2021 | 52 | 3.59 | −0.03 | −0.02 | −0.01 | 0.63 | 7.46 | 32% |
| | B4 | 19 May 2021 to 5 July 2021 | 32 | 3.57 | 0.00 | 0.01 | 0.01 | 0.41 | 8.83 | 14% |
| | B5 | 22 July 2021 to 25 October 2021 | 70 | 3.73 | 0.00 | 0.00 | 0.00 | 0.17 | 12.72 | 25% |
| GRN | B1 | 8 October 2019 to 20 December 2019 | 54 | 3.94 | −5.53 | 5.53 | 0.05 | 0.23 | 8.24 | 19% |
| | B2 | 11March 2020 to 9 April 2020 | 22 | 3.83 | −0.11 | −0.01 | −0.05 | 0.78 | 7.19 | 42% |
| | B3 | 4 May 2020 to 9 June 2021 | 310 | 3.19 | −0.02 | −0.01 | 0.00 | 0.45 | 10.98 | 220% |
| | B4 | 11 June 2021 to 1 August 2021 | 68 | 3.26 | −0.01 | 0.00 | 0.00 | 0.75 | 10.17 | 23% |
| | B5 | 19 October 2021 to 11 November 2021 | 20 | 3.37 | −0.12 | −0.01 | 0.01 | 0.58 | 10.68 | 15% |

**Table 2.** *Cont.*

| Corporate/ Index | Bubble | Time | $t_c$ | A | B | $C_1$ | $C_2$ | $\beta$ | $\omega$ | DD (%) |
|---|---|---|---|---|---|---|---|---|---|---|
| SOLCARBT | B1 | 18 March 2020 to 14 July 2020 | 89 | 4.91 | −0.13 | 0.01 | 0.01 | 0.47 | 7.26 | 93% |
| | B2 | 28 October 2020 to 14 May 2021 | 131 | 5.29 | −0.02 | 0.00 | 0.00 | 0.66 | 11.79 | 143% |
| | B3 | 23 July 2021 to 5 October 2021 | 59 | 5.72 | 0.00 | 0.00 | 0.00 | 0.65 | 12.68 | 27% |

Note: The table above depicts the coefficients of the speculative bubbles in KRBN, GRN ETFs, and SOLCARBT index standing as proxy of global carbon credit trading, within the COVID-19 outbreak period (10 September 2019 to 11 November 2021) (please see Appendix A).

**Table 3.** Event linking with extremely large LPPL Drawdowns (> 25%).

| Sr. No. | Critical Date | Drawdown | Company/Index | Events |
|---|---|---|---|---|
| 1 | 9 June 21 | 220% | GRN | Mark Carney, a former governor of the Bank of England, and Bill Winters, the chief executive of Standard Chartered have set up a task force to finalize recommendations for smaller and more permanent governance in this new carbon-offset market. |
| 2 | 14 May 21 | 143% | SOLCARBT | Climate Action Tracker estimated that climate policies implemented across the world at present, including the effect of the pandemic, will lead to a temperature rise of 2.9 °C by the end of the century |
| 3 | 14 July 20 | 93% | SOLCARBT | According to Directive 2003/87/EC, the EU ETS covers emissions from all stationary installations in EU Member States. However, emissions from stationary installations in the United Kingdom are no longer within the scope of the Union law and the EU ETS. |
| 4 | 17 April 20 | 42% | GRN | As of April 2020, Denmark, France, New Zealand, Sweden, and the UK have built on this commitment and enshrined in law a net zero CO2 emissions target into legislation, while Suriname and Bhutan are already carbon negative. In addition, 15 subnational regions, 398 cities, 786 businesses, and 16 investors have also indicated that they are working toward achieving net zero emission targets. |
| 5 | 14 May 21 | 32% | KRBN | Climate Action Tracker estimated that climate policies implemented across the world at present, including the effect of the pandemic, will lead to a temperature rise of 2.9 °C by the end of the century. |
| 6 | 17 December 20 | 28% | KRBN | More than 8.6 million future vintage allowances, cleared at $17.35/tonne as entities, could buy state-owned allowances before the end of the current compliance period. |
| 7 | 8 November 21 | 25% | KRBN | A spokesperson for Bezos' $10 billion "Bezos Earth Fund" told The Independent that the Amazon founder also "offsets all carbon emissions from his flights". In his 2021 book, "How to Avoid a Climate Disaster", Gates writes that he counteracts his non-aviation emissions by "buying offsets through a company that runs a facility that removes carbon dioxide from the air". |

Note: This table depicts seven bubble crashes (two carbon credit energy ETFs) with a drawdown of more than 25%. Further, it links those crashes with specific events (collected by the first Author from various credible sources).

We identified a total of 10 bubble crashes in the KRBN, GRN, and SOLCARBT index from 10 September 2019 to 11 November 2021 (Table 2). Interestingly, all of them exhibited a prominent LPPL signature (Filimonov and Sornette 2013). Having said that, the minimum drawdown is double (14%) that of most equities (7%) (Ghosh et al. 2021, 2020). The average of the 10 worst equity drawdowns in the S&P-500 from 1970 till 2020 was 27%[9]. The average drawdown (DD) was around 27% through various global studies (Johansen and Sornette 2010) (see representation in Figure 1). On the other hand, the average DD in the carbon credit bubble was around 54% under trying conditions (such as COVID-19), which is about two times more.

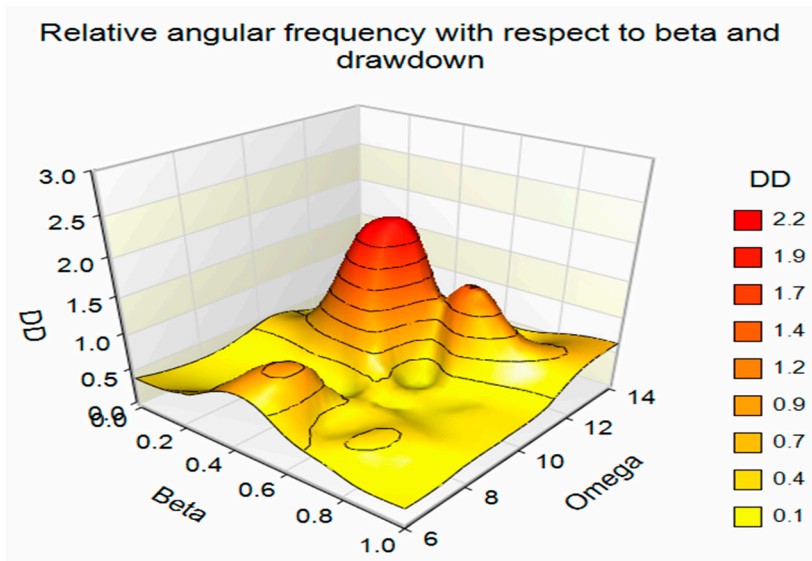

**Figure 1.** Depiction of drawdown (DD) with respect to the growth ($\beta$) and frequency ($\omega$) of the same t.

As a matter of fact, half of the carbon credit bubble has 25% or more draw-ups; just before their inevitable collapse (Table 2). Since COVID-19 showed the importance of carbon credit transfers, most European Union and North American carbon credit energy companies have been sought after by investors of all kinds. The same as above is reflected also in KRBN ETF (for North America), GRN ETF (for Europe), and SOLCARBT index prices. Furthermore, this uncanny surge in deploying funds created a repetitive, speculative bubble (Minsky moment or social bubble) in practically no time. Generally, a bubble is often associated with sharp crashes destroying investors' wealth. However, it could be a "Minsky moment", signifying a clear speculative bubble or a relatively positive Social bubble, which, in the long run, increases infrastructure spending and development (due to investment in green energy companies through carbon credit transfer) (Giorgis et al. 2021). On one hand, the social bubble hypothesis describes these bubbles as strictly non-destructive in nature. Contrary to that, the Minsky moment identifies these bubbles as a signal of financial system instability. Where a social bubble emphasizes the enhancement of economic activity and infrastructural spending, the Minsky moment would indicate real financial schemes turning into a "Ponzi scheme" owing to over-enthusiasm and excessive positive feedback. Fundamentally positive feedback loops move in a repetitive manner to generate an unprecedented growth in valuation very quickly. Terming the excess divergence from fair valuation as "Ponzi" could well be unfair, argues Davidson (Davidson 2008). Moreover, it is difficult to determine the exact transition point of a speculative bubble turning into a "Ponzi" profile (Galbraith and Sastre 2009). These critical references added a whole new dimension to the study. As was to be expected high-net-worth investors and private equity funds realized the importance of carbon credit transfer during COVID-19, which did not take much time to get translated into action. The same impact became visible through KRBN ETF, GRN ETF, and the SOLCARBT index, as they track global carbon credit indices.

Figure 1 depicts the bubble build-up with respect to certain stylized facts, which emerged from this empirical analysis.

We have found that bubble build-up in carbon credit transfer ETFs (read as KRBN and GRN) and index (read as SOLCARBT) have coefficients of exponential growth, or $\beta$ to be between 0.14 and 0.91 and coefficients of angular frequency during such growth, or $\omega$ to be between 13.04 and 6.26. Hence, future bubble build-up can be tracked in advance with these stylized facts.

About 55% of our sample exhibited at least one carbon credit bubble with DD $\geq$ 25% (Tables 2 and 3). Additionally, we observe an intuitive association between the largest (DD $\geq$ 25%) speculative bubbles for three clear reasons (Table 3): First, the climate tracker has a direct role. Second, energy policies, carbon offset governance, and global declarations around the same subject are fundamental. Third, influencers were followed by a large (crowd) herd in this regard. Basically, carbon credit or carbon allowances are issued in a decreasing number, thus demand surges naturally. Less polluting companies find it beneficial to sell, whereas high polluting companies find it worth buying, as the transition to technology takes time. Carbon credit or carbon allowances are worth a further increase, as they work as a near-perfect hedge against other asset classes and credit downgrades. A bubble is logical to happen. Carbon tax can potentially restrict these bubbles. It is clear that these bubbles are not Minsky, as their underlying asset is not "Ponzi". These are born purely out of influencer euphoria within social circles of influence (read as large corporations and Hedge Funds) (Papathanasiou et al. 2021). Far too many positives constructed a positive spiral of feedback, thus propelling these bubbles. As a result, infrastructure investments do increase, especially in line with low carbon guidelines.

## 5. Conclusions

According to Minsky (2016) "the behavior of our economy depends upon the base of investment. In a capitalist economy the valuation that is placed upon capital assets, which determines current investment, and the ability to fulfill contractual commitments, which determines financing possibilities, depend critically upon the pace of gross profits. Gross profit, in turn, is largely determined by investment. Thus the ability to debt finance new investment depends upon expectations that future investment will be high enough in order for future cash flows to be large enough for the repayment or refinancing of debts that are issued today". Furthermore, "innovations in financial practices are a feature of our economy, especially when things go well. New institutions, such as Real Estate Investment Trusts (REITs), and new instruments, such as negotiable Certificates of Deposits, are developed and old instruments, such as commercial paper, increase in volume and find new uses. However, each new instrument and expanded use of old instruments increases the amount of financing that is available and which can be used for financing of activities and taking positions in inherited assets. Increased availability of finance raises the prices of assets compared with the prices of current output, and this leads to an increase in investment. The quantity of relevant money in an economy in which money conforms to Keynes's definition is endogenously determined".

In this study, we have investigated the carbon credit bubble behavior in the ETF prices of KRBN, GRN (Global Carbon Credit tracking ETFs), and the SOLCARBT index during the COVID-19 pandemic period. The covered period is from 10th September 2019 to 11th November 2021, where daily closing prices were taken into consideration. In this study, we adopt the log-periodic power law model (LPPL) methodology. Over the past decade, the LPPL model has been widely used for detecting bubbles and crashes in various markets (Brée and Joseph 2013; Zhou et al. 2018).

Our analysis led us to the following conclusions: First of all, the presence of 13 carbon credit bubbles was found during this period. All the carbon credit bubbles followed the LPPL signature when re-calibrated using the Filimonov and Sornette (2013) LPPL equation. Secondly, the average drawdown emerged as two times that of S&P-500 listed stocks (54% vs. 27%) under stressed conditions such as those of COVID-19. This emerged as a

stylized fact. Thirdly, however, these bubbles are not Minsky, since their underlying assets are certainly not "Ponzi"; they are rather social bubbles with a defined positive impact, propelled by the newfound interest in carbon credit trading. Finally, the stylized facts obtained from our empirical analysis would assist in predicting carbon credit bubbles in the future. Thus, this study would most certainly assist policymakers, the industry, and academia alike.

According to a recent IMF blog post (Adrian et al. 2022), the economic and health benefits of phasing out coal are significant enough that we should push harder for global agreements that unleash the potential power of capital markets. "If innovative financing packages could incentivize advanced, emerging and developing economies alike to end the use of fuel' use, the net social gains from such an agreement would be enormous". In this regard, we can see carbon credit bubbles as a learning platform that helps capital markets support the transition of green finance and take a leading role in creating more sustainable, creative, and inclusive economies.

In this regard, we view the results of our study as important for both the economy and climate policy, as it would most certainly assist policymakers, the industry, and academia alike. Carbon credit bubbles can be viewed as a learning platform that helps capital markets support the transition of green finance and take a leading role in creating more sustainable, creative, and inclusive economies.

**Author Contributions:** Conceptualization, B.G.; methodology, B.G.; software, B.G.; validation, B.G. and S.P.; formal analysis, B.G.; investigation, B.G. and K.G.; resources, B.G.; data curation, B.G.; writing—original draft preparation, B.G. and V.D.; writing—review and editing, S.P. and K.G.; visualization, B.G. and K.G.; supervision, B.G. and S.P.; project administration, B.G., S.P. and K.G. All authors have read and agreed to the published version of the manuscript.

**Funding:** This research received no external funding.

**Institutional Review Board Statement:** Not applicable.

**Informed Consent Statement:** Not applicable.

**Data Availability Statement:** Not applicable.

**Conflicts of Interest:** The authors declare no conflict of interest.

**Appendix A**

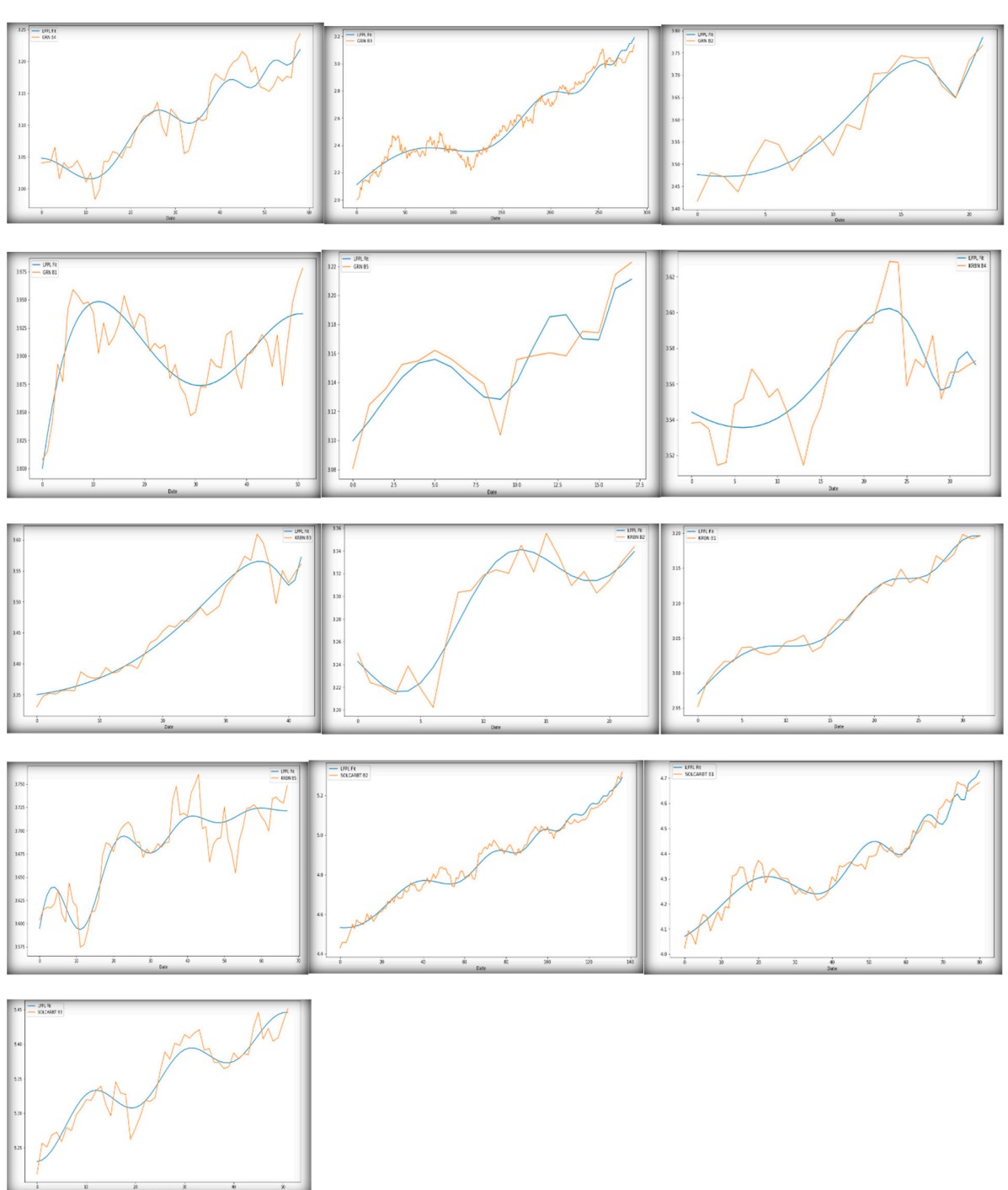

**Figure A1.** Proof of LPPL Signatures. The Bubbles in KRBN, GRN ETFs, and SOLCARBT Index Standing as Proxy of Global Carbon Credit Trading, within the COVID-19 Out-Break Period (10 September 2019 to 11 November 2021).

**Notes**

[1]   John Kerry Backs New Carbon-Price ETF in Climate Change Fight. Available online: https://www.bloomberg.com/news/articles/2020-07-30/john-kerry-backs-new-carbon-price-etf-in-climate-change-fight#xj4y7vzkg (accessed on 6 August 2022).

[2]   Niche asset nears mainstream as investors warm to EU carbon market. Available online: https://www.ft.com/content/4535374d-11ab-31aa-9908-b7c0ac2ee266 (accessed on 10 June 2022).

3    Carbon Pricing Dashboard. Available online: https://carbonpricingdashboard.worldbank.org/ (accessed on 10 June 2022).

4    Wall Street's Favorite Climate Solution Is Mired in Disagreements. Available online: https://www.bloomberg.com/news/features/2021-06-02/carbon-offsets-new-100-billion-market-faces-disputes-over-trading-rules (accessed on 10 June 2022).

5    United Nations Climate Change (n.d.). Emissions Trading. Available online: https://unfccc.int/process/the-kyoto-protocol/mechanisms/emissions-trading (accessed on 10 June 2022).

6    World Bank (n.d.). What is Carbon Pricing? Available online: https://www.worldbank.org/en/programs/pricing-carbo. (accessed on 10 June 2022).

7    Refinitive. Carbon Market Year in Review 2020. 2021. Available online: https://www.refinitiv.com/content/dam/marketing/en_us/documents/reports/carbon-market-year-in-review-2020.pdf. (accessed on 10 June 2022).

8    *Drawdown is the cumulative loss from one local maximum to the immediate next minimum; a size that is above the threshold 'ε'.*

9    Historical Drawdowns for Global Equity Portfolios. Available online: https://steadyoptions.com/articles/historical-drawdowns-for-global-equity-portfolios-r595/ (accessed on 10 June 2022).

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
