# Peer review of "Bubble in Carbon Credits during COVID-19: Financial Instability or Positive Impact (“Minsky” or “Social”)?"

_jrfm, doi:10.3390/jrfm15080367_

Round 1
Reviewer 1 Report
The topic of the paper is up-to-date.
The research flow is clear. The used method is appropriate. The presentation of the results is clear.
Some comments for improving the understandability:
The second part of title of the paper is not clear. Maybe the professionals are familiar with Minsky theory, but I am sure the wide readers are not.
Also there are references to “Minsky” and “Ponzi” in the Abstract, but these are not explained.
I have the same problem with the introduction. What is the relationship between Minsky theory and the financial market? It has to be explained. What is the “Ponzi”? It should be explaines in the introduction as well. They are only mentioned and explained for the first time in the Literature review only.
Regarding the Literature review
The used literatures are relevant. The references are mostly correct.
The No17 literature source in text citation is not correct. On the page 6, the year after the name “Haken” is missing (1975).
Johansen (19): there are two in text citation on page 6. Pls use only one.
Lohmann (30): in the in text citation pls correct the year of publication (from 2011 to 2010)
There is no in text citation on 42 literature.
The citation form for 44 literature source on the 2nd page is incorrect. The year is missing. (2018)
(Rickeet al. 2018) in text reference is not listed in the References (List)
Formal point of view
There are a lot of abbreviations in the Abstract. ETF is not detailed.
I cannot open the link of the footnote 1 on the page 2. It has to be deleted.
There is “Appendix. Proof of LPPL Signatures” with 13 graphs at the end of paper. There is no reference to the Appendix in the paper and what about the graphs it is not explained. Moreover they are not visible.
Author Response
Please see tha attached file

Reviewer 2 Report
The abstract should be adopted to journal requirements. There is a lack of the paper aim. The paper goal should be transparent and directly written.
The abstract should encourage reading the manuscript. The abstract should be better organized.
Introduction - the aim is written but should be the same as in the abstract. (See also the previous remark.)
What is the research's importance? Why do the Authors take into consideration such a subject? Especially that, the Authors do not refer (too much) to ETFs and Carbon Credit ETFs. What is new in the case of the proposed study and approach?
The literature review should be better organized also, including remarks to ETF.
Methodology and data - why do the Authors use such kinds of methods? What are the advantages of the proposed approach?
Results and conclusions should be depth and constructive. What is the importance of the results for the economy or climate policy?
Author Response
Please see tha attached file
